# A Novel Magnetic Resonance Imaging-Compatible Titanium Alloy Wire-Reinforced Endotracheal Tube

**DOI:** 10.3390/ma15165632

**Published:** 2022-08-16

**Authors:** Bikei Ryu, Yoshikazu Okada, Nobuko Fujita, Yasuko Nagasaka

**Affiliations:** 1Department of Neurosurgery, St. Luke’s International Hospital, 9-1 Akashi-cho, Chuo-ku, Tokyo 104-8560, Japan; 2Department of Neuroendovascular Therapy, St. Luke’s International Hospital, 9-1 Akashi-cho, Chuo-ku, Tokyo 104-8560, Japan; 3Department of Anesthesia, St. Luke’s International Hospital, 9-1 Akashi-cho, Chuo-ku, Tokyo 104-8560, Japan; 4Department of Anesthesiology, Tokyo Women’s Medical University, 8-1 Kawada-cho, Shinjuku-ku, Tokyo 162-8666, Japan

**Keywords:** alloys, anesthesia, intubation, magnetic resonance imaging, neurosurgery

## Abstract

Reinforced endotracheal tubes (ET) are advantageous in preventing tube obstruction and kinking by procedural compression during neurosurgeries. However, the standard reinforced ET contains an embedded stainless steel (SS) helical wire, which produces artifacts and heat during magnetic resonance imaging (MRI). Therefore, MRI is not indicated in the presence of a reinforced ET containing SS. To overcome this challenge, we developed an MRI-compatible titanium (Ti) reinforced ET. A newly developed Ti alloy helical wire was inserted in a reinforced ET. Here, we report our first clinical experience with six patients who underwent neurosurgery intubated with this Ti-alloy-reinforced ET. The Ti-alloy-reinforced ET was used in six patients requiring reinforced ET intubation. It was clearly delineated on radiography, and metal artifacts were small on computed tomography. Patients intubated with the Ti-alloy-reinforced ET could safely undergo MRI under sedation. MR images without remarkable susceptibility artifacts were obtained without noted adverse effects. We invented a novel Ti-alloy-reinforced ET. This device allows clinical use during MRI because it is less susceptible to artifacts in high magnetic fields.

## 1. Introduction

Spiral embedded endotracheal tube (ET) is known as reinforced ET to minimize the probability of obstruction or compression of the ET during the surgery. A reinforced ET is designed to make the tube flexible and resist kinking with embedding concentric metal wire rings inside the tube. In head-and-neck surgery and spinal surgery, including neurosurgery, the use of a reinforced ET may be considered to prevent ET obstruction and kinking by mechanical compression due to lesions or patient posture during surgery requiring excessive neck flexion or twisting under general anesthesia, such as in the prone and lateral positions. Especially in the neurosurgical field, patients may require prolonged intubation management because of prolonged unconsciousness, and magnetic resonance imaging (MRI) is in high demand for lesion examination during tracheal intubation. Furthermore, in recent years, intraoperative MRI has become a popular technique in neurosurgery, and the ability to perform intraoperative MRI under general anesthesia during intubation is a marked advantage [1,2].

However, conventional reinforced ETs contain stainless steel (SS) wires wrapped inside them for kink resistance. MRI during intubation is contraindicated because of metal artifacts or heat generation due to the SS magnetic material, which has been a challenge. Thus, we hypothesized that an ET containing wires made of titanium (Ti), which produces very few susceptibility artifacts on MRI [3,4,5], would be MRI compatible. Our work on the development of a Ti-alloy wire that produces markedly few metal artifacts on MRI has led to the development of an MRI-compatible reinforced ET with a Ti-alloy wire.

In this study, the metal artifacts of invented Ti alloy helical wire and conventional metal wire were evaluated in comparison using MRI. Furthermore, MRI compatibility of the Ti-rET was evaluated in vitro and ex vivo. The Ti-alloy-reinforced ET (Ti-rET)’s MRI-compatibility could be ensured through the use of paramagnetic material instead of the conventional SS wire. We also report our clinical experience with the newly developed Ti-rET; we also illustrate its efficacy and the fact that it produces few metal artifacts even in high-magnetic-field MRI. There has never been a reinforced ET that has been approved as a medical device capable of MRI imaging. We were able to achieve MRI compatibility by changing the SS wire used in conventional reinforced ET to a Ti alloy with less metal artifacts. This product has already been approved as a medical device in Japan and has been developed to the point where it can be used for patients.

## 2. Materials and Methods

All procedures performed in this series involving human participants were in accordance with the ethical standards of the institution (2 June 2022, No. 22-R019, St. Luke’s International Hospital, Tokyo, Japan) and with the 1964 Helsinki declaration and its later amendments or comparable ethical standards. This retrospective study included six consecutive patients between April 2018 and January 2019. The requirement for written informed consent was waived owing to the retrospective design of the study.

### 2.1. Titanium-Alloy Wire-Reinforced Endotracheal Tube (Ti-rET)

A newly developed Ti-alloy (Ti-6Al-4V ELI, Fuji Systems Inc., Kanagawa, Japan) helical wire-filled reinforced endotracheal tube (0200122-0200136, Fuji Systems Inc.) was created. This Ti-rET tube was made by adding a Ti alloy helical wire and silicon cuff to a silicone tube (Figure 1). This Ti alloy helical wire is elliptical cross-section and embedded inside the tube wall along its entire length. It was designed to increase tube flexibility and kinking resistance independent of the patient’s position. This product has been verified that the temperature rise does not occur after 15 min of MRI imaging under 3.0-Tesla conditions in accordance with the American Society for Testing and Materials (ASTM), and that the maximum artifact is 3.5 mm for the spin echo and 10.3 mm for the gradient echo MRI protocol (ASTM F2182-11a and ASTM F2119-07).

### 2.2. Ex Vivo Metal Wire Artifact Evaluation on MRI

In vitro metal-wire MRI was performed to evaluate metal-wire artifacts on MRI. We fabricated four types of fine metal wires with lengths of 10 cm: pure Ti (Ø 0.1 mm), nickel-titanium alloy (Ni-Ti, Ø 0.1 mm), SS (SUS304, Ø 0.1 mm), and Ti alloy (Ti-6Al-4V ELI, Fuji Systems Inc., Ø 0.21 mm) used in commercially available Ti-rET (Fuji Systems Inc.). Each metal wire was balled-up and placed into the tip of 2.5-mL polypropylene syringes (Figure 2a). Each syringe was installed in a polyethylene terephthalate cylinder filled with water to mimic the internal body environment. Each cylinder was scanned using 3.0-Tesla MRI (Figure 2). Ex vivo MRI evaluation of commercially available Ti-rETs was also performed. The Ti-rET fixed to the phantom model was scanned using 3.0-Tesla MRI.

### 2.3. Imaging

3.0-Tesla MRI protocols are summarized (Discovery MR750w, GE Healthcare, Milwaukee, WI, USA). T1-weighted image (WI), T2-WI, diffusion weighted image (DWI), were performed for the metal wire artifacts. The T1-sagittal isotropic CUBE parameters (fast spin echo 3D) were: slice thickness, 1.2 mm; matrix size, 320 × 256; freq field of view (FOV), 25.0 cm; phase FOV, 22.5 cm; repetition/echo time, 600/10 ms; echo train length, 20. The T2-axial parameters (fast spin echo) were: slice thickness, 6.0 mm; matrix size, 320 × 224; freq FOV, 22.0 cm; repetition/echo time, 13,000/135 ms. The DWI-axial parameters were: slice thickness, 6.0 mm; b = 1000; freq FOV, 22; phase FOV, 1.0; repetition/echo time, 7000/78 ms. A head 24-channel coil was used. The B0 field direction is 3T, and the readout direction is AP.

### 2.4. Intubation Using the Titanium-Alloy Wire-Reinforced Endotracheal Tube

This retrospective study included six consecutive patients who underwent neurosurgical operations requiring intubation using the Ti-rET at St. Luke’s International Hospital, Tokyo, Japan, between April 2018 and April 2019. We used our Ti-rET for ventilation management during surgery and perioperative management. Intubation was performed by anesthesiologists or emergency physicians using a Macintosh laryngoscope (Welch Allyn Inc., New York, NY, USA) or McGrath (McGrath Series 5, Aircraft Medical, Edinburgh, UK). In order to analyze the ease of handling for Ti-rET, the number of attempts for successful intubation, types of laryngoscope, and the use of other assistive procedures were recorded.

The Ti-rET was used in six patients requiring reinforced ET intubation for ventilation management (five men and one woman). The mean age of the patients was 71.8 years, ranging from 47 to 73 years. The Ti-rET was used in the following surgeries: cervical arteriovenous shunt obliteration, cervical laminoplasty for cervical cord injury, cervical tumor removal, nidus removal for brain arteriovenous malformation, and two intracranial tumor removals. The procedures were performed with the patient in the supine, prone, or lateral position under general anesthesia.

## 3. Results

### 3.1. Titanium Alloy Wire-Reinforced Endotracheal Tube

Remarkable susceptibility artifacts were not observed with the Ti alloy (Ti-6Al-4V ELI), pure Ti, and Ni-Ti wires on ex vivo MRI (Figure 2 and Figure 3). In the comparison between the four types of metal wires, the SS (SUS304) wire produced remarkable metal artifacts compared to the other metal wires (Figure 2 and Figure 3). In the ex vivo imaging evaluation of the commercially available Ti-rET (Fuji Systems Inc.), few artifacts were observed on routine 3.0-Tesla MRI (Figure 4).

### 3.2. Neurosurgery with Intubation Using the Titanium-Alloy Wire-Reinforced Endotracheal Tube

All patients were intubated with a single intubation procedure. A Mcintosh laryngoscope was used in 5 cases, except for the case in which McGrath was used for preoperative Mallampati classification of Class III. No other assistive devices for endotracheal intubation were used during intubation. The Ti-rET was clearly delineated on radiography, and metal artifacts were small on computed tomography (CT) in the patients intubated with the Ti-rET. In addition, patients intubated with the Ti-rET could undergo MRI under sedation without adverse events in airway and respiratory management, and clear MR images without remarkable susceptibility artifacts were obtained. We observed no complications related to the Ti-rET in any of these intubations.

A representative image of a patient undergoing endotracheal intubation using the Ti-rET is shown in Figure 5. The patient was a 66-year-old man who was transferred to our hospital because of a motorcycle accident. He showed tetraparesis and respiratory failure due to severe cervical spinal injury. An in-line stabilization was applied while intubating the patient with a Ti-rET. Under the general anesthesia, we performed C3 laminectomy and C4–6 laminoplasty for decompression in the prone position. Ti-rET provides minimum artifact while performing CT and MRI examinations, and images of severe cervical spinal cord injury with severe cervical canal stenosis were obtained (Figure 5a–d).

## 4. Discussion

Here, we reported the effectiveness of a novel MRI-compatible Ti-rET for use in neurosurgery and perioperative respiratory management. We identified fewer metal artifacts with use of our ET in both ex vivo and clinical imaging. To our knowledge, this is the first reinforced ET using a Ti alloy wire, and it produces few adverse effects on MRI. Ex vivo evaluation showed that the SS wire used in conventional reinforced ETs produced substantially high metal artifacts on MRI, whereas no remarkable metal artifacts were observed with the Ti alloy wire. Based on these results, we conclude that Ti alloy wires are safer for MRI at static field strengths of 3.0-Tesla than SS wires. By adding a helical Ti alloy wire inside the entire circumference of the tube, a 3.0-Tesla MRI-compatible ET was developed while maintaining the conventional shape and mechanism to prevent tube lumen obstruction due to kinking or external compression.

The development of ET has been closely related to advances in general anesthesia and surgery. Various reinforced ETs have been designed for flexibility and to resist kinking due to surgical positioning and manipulation of the head and neck during patient positioning [6]. Until the introduction of our Ti-rET using a Ti alloy helical wire, reinforced ETs were conventionally manufactured using an embedded SS helical wire [6], producing severe artifacts and heat on MRI. Therefore, MRI examination is not indicated in the presence of a reinforced ETs containing SS. To overcome this challenge, materials that are less magnetic and processable are required. However, embedding of small helical wires in ETs has not been well established because the best shape and size of the coil remain unclear. In our ex vivo study, we demonstrated less significant artifacts with all types of Ti materials, including Ti alloys, on MRI. Concerning heat generation in a magnetic field, the Ti alloy wire used for Ti-rET showed an increase of less than 2 °C (unpublished data).

Ferromagnetic materials (Fe, Mg, Ni, etc.) strongly affect and become magnetized in a magnetic field [7], paramagnetic materials (Ti, Al, etc.) are magnetized, slightly increasing the strength of the magnetic field [7]. Thus, Ti alloys are considered preferable for MR-compatible medical applications to reduce MRI artifacts because of the low susceptibility of the material even under 3.0-Tesla magnetic fields [3,4,5,8,9]. The Ti alloy, Ti-6Al-4V ELI, used in our Ti-rET has high strength, excellent corrosion resistance, superior low-temperature strength, biocompatibility, and versatility and is a representative non-ferromagnetic material for medical use, especially for applications in the neurosurgical field, such as in cerebral aneurysm clipping [3,4,9,10,11].

Ti-6Al-4V ELI used for our Ti-rET is an alpha-beta alloy containing 6% Al and 4% V with excellent heat-treatability and a good strength and ductility balance. This alloy exhibits an excellent combination of corrosion resistance, strength biocompatibility and mechanical resistance [12,13]. Ti-6Al-4V ELI has already been registered as a metal for medical materials in ASTM standard (F136). The reinforced ET, which is the objective of this study, was conventionally made using stainless steel wire with an extremely high elastic module strength and thus has kink resistance that prevents tube obstruction. Therefore, our newly developed Ti-rET also required flexibility, high kink resistance, and strength to retain the ET lumen. To achieve these goals, we chose Ti-6Al-4V ELI. Although various new Ti alloys have been developed, the recent development of implant biomaterials tends to seek low elastic modules as well as fatigue strength [14]. Therefore, we considered Ti-6Al-4V to be more suitable in terms of mechanical properties for a good strength, as indicated by the elastic module or Young’s modulus [13,14,15].

The ability to perform MRI safely during Ti-rET intubation is considered a significant advantage in patient management, although susceptibility artifacts that appear around materials on MRI examinations impair lesion assessment during or after surgery. In recent years, the effectiveness of intraoperative MRI in neurosurgery has been widely recognized, and intraoperative image navigation and evaluation of the extent of brain tumor removal require intraoperative MRI [1,2]. Intraoperative MRI should be used with materials that few susceptibility artifacts and adverse effects [2]. Moreover, in awake surgery for brain tumor resection in an intraoperative MRI environment, the patient’s airway is secured by means of supraglottic airways or endotracheal intubation without a metal-reinforced shaft for safe brain tumor resection [1]. However, ET obstruction is secondary to external forces that cause tube kinking by the patient’s neck posture or compression by biting due to shallow sedation [6,16,17,18]. Tube obstruction or kinking of the ET can be life-threatening, and the use of a reinforced ET may be preferred in such cases. Note that a reinforced ET can lead to blockage if it collapses completely; it cannot return to its normal shape and must be changed. Therefore, risk avoidance, such as inserting bite blocks for prevention of the tube biting, is necessary.

This study had some limitations. First, no trial production of reinforced ET with other Ti alloy wires has been conducted. Second, there is no product comparison data with other titanium alloys. Although the metal artifacts of our Ti-rET is small enough for clinical use, we believe that further investigation and product development with other alloys will be necessary to develop better products in the future. The strength target to obtain kink resistance, which is the original purpose of reinforced ET, cannot be neglected. Development of new titanium alloys with superior mechanical properties will be considered to further increase the kink resistance of MRI compatible reinforced ET.

## 5. Conclusions

We invented a novel Ti-alloy-reinforced ET. This device allows clinical use during MRI because it is less susceptible to artifacts in high magnetic fields.

## Figures and Tables

**Figure 1 materials-15-05632-f001:**
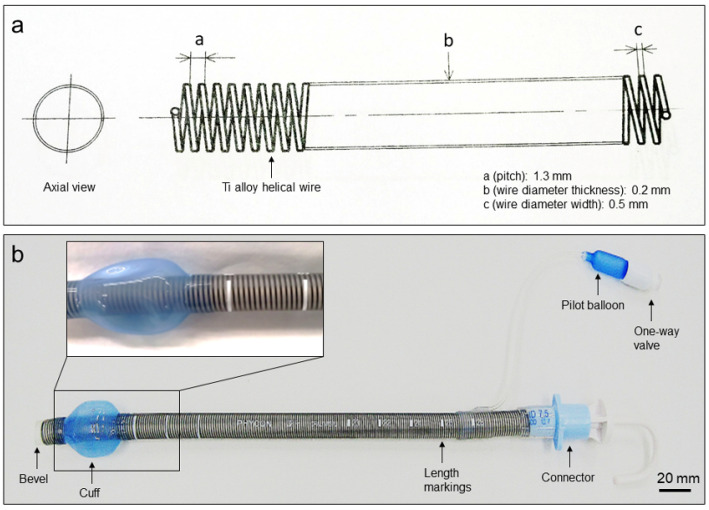
The novel MRI-compatible Ti alloy helical wire-reinforced endotracheal tube (Ti-rET). (**a**) Blueprint of the Ti-rET. a, pitch 1.3 mm; b, wire diameter thickness 0.2 mm; c, wire diameter width 0.5 mm (**b**) An overview of the Ti-rET.

**Figure 2 materials-15-05632-f002:**
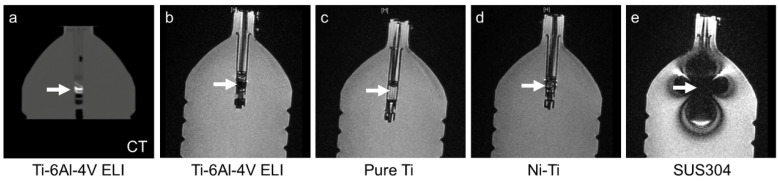
Comparison of the images of metal wires in the phantom on MRI (3.0-Tesla). (**a**) CT image of Ti alloy (Ti-6Al-4V ELI, Ø 0.21 mm) used in our reinforced endotracheal tube. To evaluate the susceptibility artifacts of various metal wires, coronal T1-weighted MRI of helical metal wires (**b**–**e**) was performed: (**b**) Ti alloy (Ti-6Al-4V ELI, Ø 0.21 mm) used in our reinforced endotracheal tube, (**c**) pure Ti (Ø 0.1 mm), (**d**) Ni-Ti alloy (Ni-Ti, Ø 0.1 mm), and (**e**) SS (SUS304, Ø 0.1 mm). The white arrows show the helical wires inside the tip of the 2.5-mL polypropylene syringes.

**Figure 3 materials-15-05632-f003:**
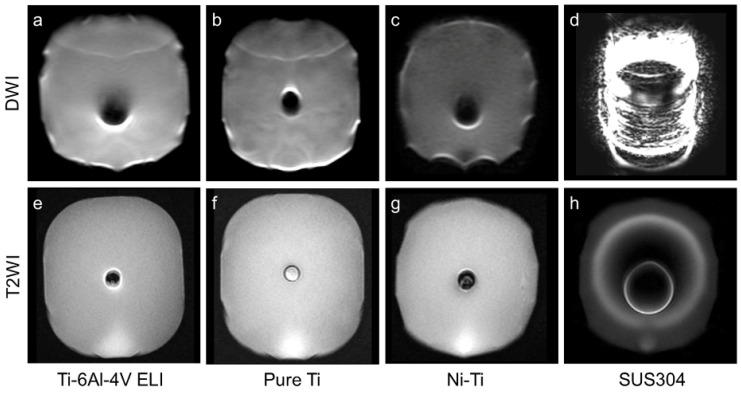
Comparison of the images of metal wires in the phantom on MRI (3.0-Tesla). To evaluate the susceptibility artifacts of various metal wires, axial DWI (**a**–**d**) and T2-weighted (**e**–**h**) MRI of helical metal wires was performed: (**a**,**e**) Ti alloy (Ti-6Al-4V ELI, Ø 0.21 mm) used in our reinforced endotracheal tube, (**b**,**f**) pure Ti (Ø 0.1 mm), (**c**,**g**) Ni-Ti alloy (Ni-Ti, Ø 0.1 mm), and (**d**,**h**) SS (SUS304, Ø 0.1 mm).

**Figure 4 materials-15-05632-f004:**
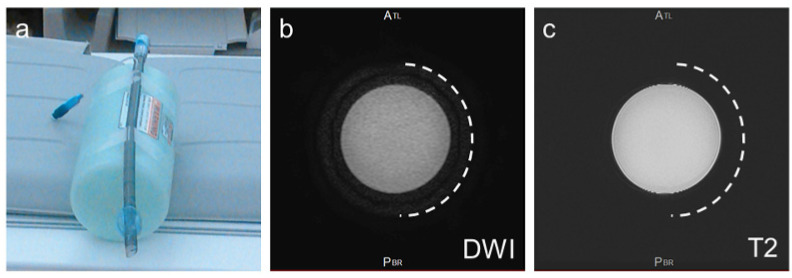
Ex vivo MRI evaluation of the commercially available Ti alloy helical wire-reinforced endotracheal tube (Ti-rET). (**a**) The Ti-rET fixed to the phantom model was scanned with 3.0-Tesla MRI. Axial DW image (**b**) and T2-weighted image (**c**) of the endotracheal tube. The semi-circle dotted lines indicate the outline of the tube.

**Figure 5 materials-15-05632-f005:**
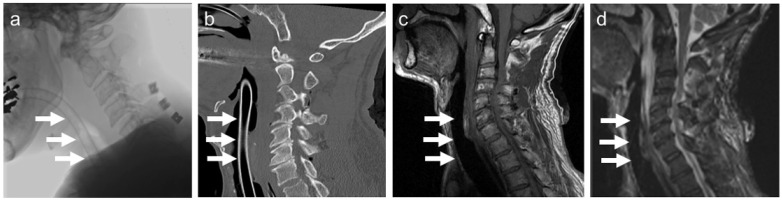
Clinical use of the Ti alloy helical wire-reinforced endotracheal tube (Ti-rET). (**a**) Radiographic image, (**b**) sagittal computed tomography image, (**c**) sagittal T1-weighted MR image, and (**d**) sagittal T2-weighted MR image. The white arrows show the intubated Ti-rET.

## Data Availability

All relevant data supporting the results of the present study are included within the article and can be obtained from the corresponding author on reasonable request.

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
