# Peer review of "A Novel Magnetic Resonance Imaging-Compatible Titanium Alloy Wire-Reinforced Endotracheal Tube"

_materials, 2022, doi:10.3390/ma15165632_

Round 1
Reviewer 1 Report
In the study described in the manuscript, an endotracheal tube (ET) is investigated that incorporates a helical Ti alloy wire for increasing the mechanical stiffness of the tube. The tube was manufactured in house. It was investigated by planar x-ray imaging, CT, and MRI, both in a phantom and in vivo (in six patients). Qualitatively, the acquired images show less MR image artefacts as compared to an endotracheal tube reinforced with a stainless steel wire. It is concluded that the presented device is "safe for clinical use".
----MAJOR COMMENTS----------------
1) The introduction is rather short. The necessity of using metal for ET reinforcement at all should have been argued for in more detail.
2) Ethics (p50ff.): (i) in my opinion, the statement "requirement for written informed consent was waived owing to the retrospective design of the study" is not acceptable. Patients do not waive a right. Was the specific study design approved by the local ethics committee? How was the patients' right associated with their own data be granted? (ii) The authors declare to follow the STROBE guidelines mentioned (reference missing) but do not follow the STROBE policy to avoid the term "retrospective" or to exactly define it by adding detailed information.
3) l73: more information should be provided on the Ti alloy. The term "Ti-6Al-4F ELI" can be deciphered by a web search but still the material should be exactly specified. Was the alloy produced in-house? If not, where was it purchased from? What does "our Ti-rET" mean? Here and sometimes in the following (e.g., l113), it is unclear which device is meant -- since a commercial Ti-reinforced ET is also considered if I understood it correctly.
4) l89, "Supplementary Material": I found no supplementary material.
5) l89 ff.: Was any k-space segmentation used for 3d fast spin echo? Any RARE factor or so? The nominal spatial resolution is missing, it should be provided.
6) In the images, the B0 field direction and the readout direction should be specified explicitly.
7) l22, l143, "patients (...) could safely undergo": this is not sufficiently supported by data -- safety comprises more than image artefacts. Heating due to rf energy deposition was not investigated, or the results mentioned on l194 (increase of less than 2 deg C) are not shown. This data should be provided including the experimental setup.
8) Fig5: actually, one would like to see images taken before and after intubation. I understand this is difficult to achieve in clinical cases.
9) l178, "low susceptibility (...) even under high magnetic fields": do you really refer to a field dependence of magnetic susceptibility?
10) l206: why are brain surgeries an issue here? I would not expect any artefacts in brain images arising from metal in an endotracheal tube.
11) There is no quantitative analysis. Only one geometry is tested. More importantly, the effectiveness of the wire in increasing mechanical stiffness is not investigated. It is not true that (l47) the efficacy is illustrated, in my opinion. Hence, it is unclear whether the suggested device is appropriate for the suggested purpose. The equivalence in mechanical effect between stainless steel and the suggested alloy should at least be argued for. Experimental data on the mechanical properties of the complete devices would be even better.
12) The artefact level for various types of wire was assessed ex vivo with a water-filled phantom containing an amount of "balled-up" wire. However, if I am not mistaken, this is far from the situation with a typical placement of the suggested device -- in a different geometry, just within the air-filled trachea. Hence, the results of these experiments may be useful for comparing different materials as such but are not very meaningful for the intended use case.
----MINOR COMMENTS----------------
13) l45: avoid hyphenating par-amagnetic (rather para-magnetic)
14) l59: the wording reads as if the Fuji product number given refers to the wire-enforced ET. However, I understood that a commercial ET without wire was purchased and then reinforced with Ti alloy wire in-house. It should be made clearer what was purchased where and what was done in-house.
15) l62: rephrase "elliptical" to something with "elliptical cross-section" or so. The phrases "diameter thickness" and "diameter width" do not make sense from my point of view (although their meaning can be figured out from Fig1). I do not understand what "the tip ... was designed with the slant facing forward" means. What's forward, what does "slant" mean?
16) l 68: abbreviations should rather be defined in the main text than in a figure caption. They should at least not be repeated at each figure -- unless this is a journal policy.
17) l72 and throughout the manuscript: the Greek letter "phi" in the wire geometry specifications does not make sense. Is this a confusion with the diameter symbol?
18) l86, "into": rephrase
19) l91 ff., "Discovery MR750w": if this is the brand name of the MR system used, the name should not occur after "artifacts". What is "CUBE" and why does it occur after "system"? The FoV and the b-value lack a unit. Phase FoV seems to be rather a fraction than a length -- clarify.
20) l103, "Macintosh laryngoscope" should be accompanied by a manufacturer's name, the registered trademark sign at "McGrath" should be removed (as other names in the text are registered trademarks as well). The phrasing "a McGrath" should be avoided (l133). A word is missing after "successful" (l105).
21) Fig4: what does "fixed to the phantom model" mean? Wasn't the syringe placed inside the water-filled cylinder in the homebuilt cases?
22) l147, "A representative images": change to singular.
23) l154, "detailed (...) injury": wrong word
24) l164, "fewer": than in which case?
25) l171, "around": the wire helix was inside the tube, not wound around the exterior, right?
26) l174, "Although": wrong word, rephrase. "are slightly magnetized": rephrase.
27) l180, "toughness": consider rephrasing.
28) l189, "ETs": change to singular.
29) l211, "byte blocks": what is this? If bite blocks are meant, specify the approach in more detail.
30) Fig3: it would be good to see the syringe diameter for comparison here.
Reviewer 2 Report
· There are many articles of titanium alloys. The following references are recommended on Titanium group: [1] In-depth assessment of new Ti-based biocompatible materials; [2] New Titanium Alloys, Promising Materials for Medical Devices; [3] Cytocompatibility of pure metals and experimental binary titanium alloys for implant materials; [4] Ti based biomaterials, the ultimate choice for orthopaedic implants - A review
· Show the novelty of the paper compared to the literature, however there is still much work on this topic.
· Why you choose these alloys?
· In the Introduction section, the last paragraph should contain the scientific contribution and scientific hypotheses of your research. Complete, further elaborate the scientific contribution and scientific hypotheses of your research. Be explicit. In addition to the goal of the research (which was written), the novelty in the context of the scientific contribution should be pointed out. Scientific contributions should be written based on the shortcomings of previous research in the literature. In this way, the authors will better emphasize novelty and scientific soundness.
· Analyze and discuss possibilities of practical application.
· Complete the conclusions with the limitations of the proposed methodology. Also write future research.
· Generally, the quality of the writing could be improved.
Reviewer 3 Report
The paper “A novel magnetic resonance imaging-compatible titanium alloy wire-reinforced endotracheal tube” focuses on the in vivo and ex vivo efficacy of newly developed reinforced endotracheal tube of titanium alloy and the metal artifacts which it produces in high-magnetic-field MRI. In contrast to stainless steel, Ti-6Al-4V ELI has excellent mechanical, corrosion properties, and biocompatibility, making it suitable for replacing the conventional reinforced endotracheal tube. The tests used are not versatile but they are enough to fulfill the purpose set out by the authors. The introduction refers to the aim of the study. The experimental part is consistently revealed and explained while the results and discussions are understandably submitted. In my opinion, the paper may be interesting from a scientific and practical point of view.
However, I would like to recommend the publication of the manuscript in this journal after fulfilling the following recommendations:
1. The aim of the study could be re-formulated;
2. The technological methods used for the production of helical wires from different Ti alloys should be included in Section 2;
3. Since they are part of the experimental methodology, the first three paragraphs from section 3.2 should be shifted to section 2.4.
4. Conclusion section should be added to the research paper.
5. Figures 2,3 and 5 can be made larger for better visualization;
6. It is recommendable that the authors reveal the stability of the new helical wire after prolonged usage or a certain number of cleaning and sterilization procedures;
Round 2
Reviewer 2 Report
Article was improved.
Reviewer 3 Report
Although the conclusion section can be extended, the authors have carefully addressed all the reviewer's recommendations.